# Innate and adaptive immune traits are differentially affected by genetic and environmental factors

Massimo Mangino[1,2,*], Mario Roederer[3,*], Margaret H. Beddall[3], Frank O. Nestle[2,4,*] & Tim D. Spector[1,*]

The diversity and activity of leukocytes is controlled by genetic and environmental influences to maintain balanced immune responses. However, the relative contribution of environmental compared with genetic factors that affect variations in immune traits is unknown. Here we analyse 23,394 immune phenotypes in 497 adult female twins. 76% of these traits show a predominantly heritable influence, whereas 24% are mostly influenced by environment. These data highlight the importance of shared childhood environmental influences such as diet, infections or microbes in shaping immune homeostasis for monocytes, B1 cells, $\gamma\delta$ T cells and NKT cells, whereas dendritic cells, B2 cells, CD4$^+$ T and CD8$^+$ T cells are more influenced by genetics. Although leukocyte subsets are influenced by genetics and environment, adaptive immune traits are more affected by genetics, whereas innate immune traits are more affected by environment.

[1] Department of Twin Research and Genetic Epidemiology, Kings College London, London SE1 7EH, UK. [2] NIHR Biomedical Research Centre at Guy's and St Thomas' Foundation Trust, London SE1 9RT, UK. [3] ImmunoTechnology Section, Vaccine Research Center, NIAID, NIH, 40 Convent Drive, Bethesda, Maryland 20817, USA. [4] Cutaneous Medicine Unit, St John's Institute of Dermatology, King's College London, London SE1 9RT, UK. * These authors contributed equally to this work. Correspondence and requests for materials should be addressed to M.R. (email: roederer@nih.gov) or to F.O.N. (email: frank.nestle@kcl.ac.uk).

Immune protection is a remarkably balanced defence mechanism to protect the host from environmental threats and pathogens without triggering aberrant responses to self-antigens that underlie autoimmunity. The balance of a large diversity of leukocyte subpopulations is driven by genetic and environmental influences that maintain homeostasis of innate cell types (pre-programmed to respond to pathogens and cancers), naive adaptive B and T lymphocytes (comprising antigen receptors that theoretically could target any un-encountered pathogen or neo-antigen) and functionally polarized memory B and T lymphocytes (that can rapidly respond to previously encountered antigen). Many cancers, autoimmune diseases and immunodeficiencies result from aberrant homeostatic control over this panoply of cell types. To date, the genetic mechanisms and environmental factors that regulate homeostasis of cell numbers and phenotypes in the peripheral immune system are poorly understood.

Genes responsible for variation in the response to pathogens and inflammation regulation are common targets of natural selection[1]. In particular, studies have identified signatures of pathogen-mediated selection in genome-wide association study (GWAS) and numerous single-nucleotide polymorphisms associated with autoimmune diseases such as celiac disease, ulcerative colitis, type 1 diabetes, Crohn's disease and multiples sclerosis[2] have been identified. Despite this progress, few of these genetic associations have led to mechanistic insight.

Studies more than a decade ago showed that overall numbers of $CD8^+$ and $CD4^+$ T cells are under genetic influence[3]. Advances in technology enable detailed analysis of the human immune system on a genetic and phenotype level. Adopting these techniques, several studies[4–8] report heritability estimates on selected immune subtypes. Some of the studies emphasize the heritable nature of immune traits, whereas others focus on environmental influences. The differences in interpretation between such studies are mostly due to different assessment methods and/or the size of the analysed data set. With our increasing knowledge of complex disease genetics, as well as quantifiable environmental factors such as lifestyle factors and the microbiome, considerable interest exists regarding the extent to which environmental versus genetic factors influence human immune cell homeostasis.

We previously reported the discovery of 11 genetic loci affecting 19 well-defined immune traits by focussing on the analysis of the 151 most heritable immune traits per a prespecified statistical analysis plan. Here we extend that analysis to define the genetic and, uniquely afforded by the twin-based design of our study, the shared environmental influence on the variation in 23,394 robust immune phenotypes in 497 adult female twins (TwinsUK) profiled with a high-resolution deep immunophenotyping flow cytometry approach. We show that there is broad heritability of most human immune traits and, using the power of this well-balanced twin cohort, separately quantify shared versus unique environmental influences. Our study assists precision medicine by defining which human immune traits are under genetic or environmental control and, in particular, which are subject to common shared household exposures such as microbiota and diet.

## Results

### Genetic and environmental dependence of immune traits. The study cohort comprised 75 monozygotic (MZ), 170 dizygotic (DZ) twin pairs and 7 singletons. The whole data set only included females of Caucasian origin. All twin pairs were raised together. We observed no statistical differences within MZ and DZ pairs for both education level ($P = 0.09$) and personal welfare

($P = 0.2$). The MZ pairs were on average 2 years older than the DZ (MZ: $62.4 \pm 8.9$ years; DZ: $60.9 \pm 7.7$ years). The use of a twin cohort brought two major advantages to our study that are difficult to obtain in other designs. First, it allowed us to accurately identify heritable traits that were most likely to be informative using GWAS in our original analysis[8]. Second, it allows us now to identify not only heritability, but also common and shared environmental influences, on the variability of the full set of immune traits investigated. Here we report on the influence of both genetics and environment on 23,394 immune traits.

As described in Methods, we filtered our complete set of 89,051 traits to focus on those that were robust ($n = 23,394$); structured equation modelling (SEqM) analysis was performed on these traits. The full results of all this analysis are reported in Supplementary Data 1. Robust trait values are given in Supplementary Data 1 and demographic information in Supplementary Data 1.

Overall, a large fraction of traits show evidence of genetic control. As illustrated in Fig. 1a, the mean correlation of traits was substantially higher for MZ twins (mean $r = 0.61$) than for DZ twins (mean $r = 0.35$). The mean correlation in longitudinal analysis of these traits is $r = 0.90$, showing a low degree of experimental error. Using Falconer's formula to estimate heritability (Fig. 1b), the mean genetic influence was $\sim 45\%$, with almost one in eight immune traits above 80%. However, although Falconer's formula is computationally simple to perform and provides a good overview, it is an imperfect estimate for individual traits, prone to a number of potential biases and assumptions. Thus, we applied SEqM to the trait values.

For each trait, SEqM was applied with a full three component model (ACE: Additive genetic influence (A), that is, heritability; Common environmental influence (C); and unique Environment influence (E)), as well as the two-component CE or AE models; the best-fitting was chosen. A minority of the traits (24%) showed no clear genetic influence and appeared to vary primarily due to environmental factors (best-fitting model CE) with an average trait variability due to common environmental factors of 42.7% (interquartile range: 32.2–54%). The majority of the analysed traits (76%) were best explained by a model including genetic influence. AE was the best-fitting model for 54% of the analysed traits ($n = 12,717$) with a mean genetic influence of 62.2% (55.1–71.9%). ACE (that is, including shared environmental factors) was the best-fitting model for 22% of the traits (5,093) with a mean heritability estimate of 39.7% (32.8–46%).

### High frequency of heritable traits in most immune subsets. A summary of the strongly heritable signals ($>60\%$) as computed from the SEqM is shown in Fig. 1c. Highly heritable traits were present throughout the leukocyte subsets. In part, this may be expected for populations like natural killer (NK) cells, for which there is an obligatory matching of the phenotype of KIR/LIR molecules with the major histocompatibility complex class I alleles expressed by an individual. Other highly heritable traits include expression of CD39 on regulatory T (Treg) cells and the expression of CD32 on myeloid dendritic cell (DC) populations; these were fully analysed previously[8]. There were several cell types with novel findings and unexpectedly high heritability, including subsets of myeloid DCs and many CD4 T-cell subsets. Overall, major genetic determinants were found in key canonical immune cell lineages such as Treg cells, NK cells, DCs and CD4 T cells.

To explore this further, we illustrated the data in 'circle' plots that depict the relative contribution of heritable and environmental (both shared and unique) influences on the variation of each trait. These are separately shown for the 23,005 cell subset

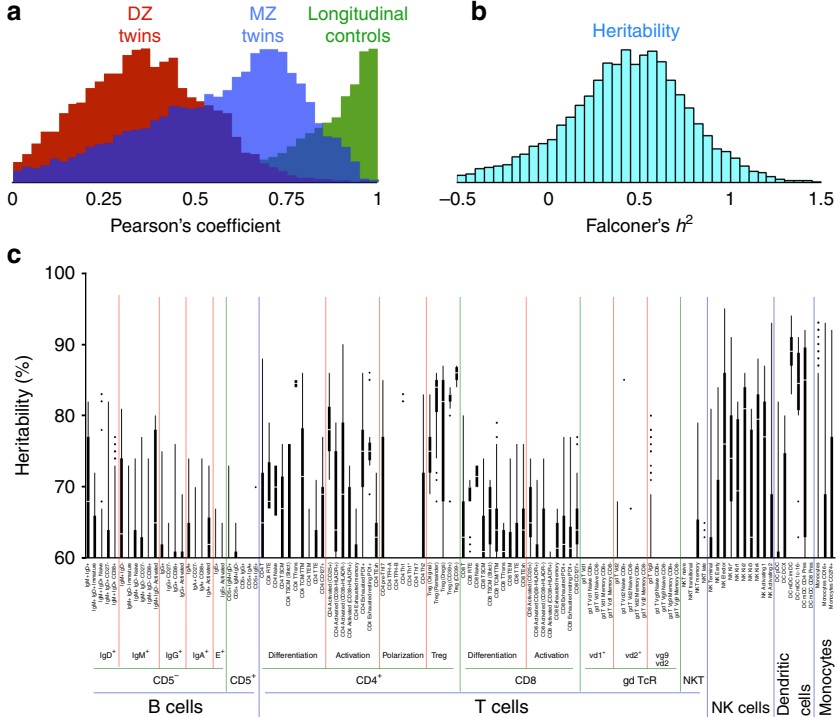

**Figure 1 | Estimating heritability of immune traits.** (**a**) Trait values from MZ twins, DZ twins or longitudinal specimens were correlated; the Pearson''s correlation coefficients for all traits are shown as histograms. (**b**) The distribution of heritability of all traits as estimated by Falconer's formula (that is, twice the difference of the Pearson's $r$ for MZ and DZ twins). (**c**) SeqM was performed on all robust traits. The traits were grouped according to cell subsets as shown under the axis. The distribution of genetic associations for traits in each category is shown as a bar chart: thick bars indicate the interquartile range (with a break at the median); thin lines indicate the 10th–90th percentiles, with dots for outliers. It is noteworthy that traits with heritabilities <60% are not shown here.

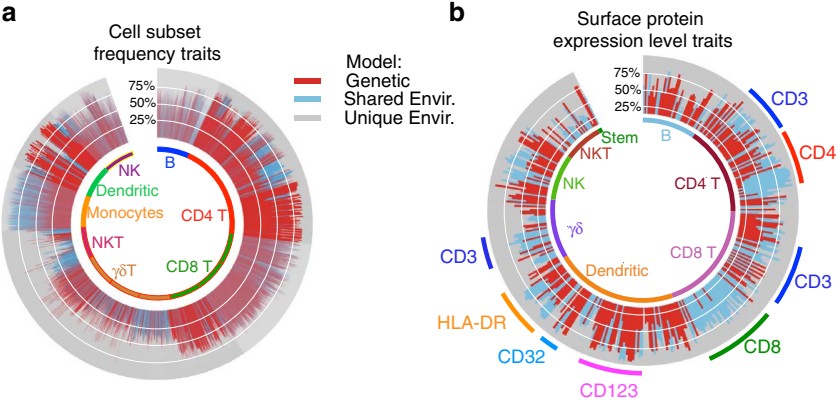

**Figure 2 | Proportion of trait variation explained by heritable or environmental influence.** Based on SeqM, the fraction of variation in a trait is calculated for heritable, shared environmental or unique environmental factors. This is shown in circle plots for CSF traits (**a**) and SPEL traits (**b**). Traits are arranged in order by the major lineage to which they belong (as indicated inside each circle); for SPEL traits, they were further ordered according to which cell surface protein was quantified (some of which are indicated outside the circle).

frequency (CSF) traits (Fig. 2a) and the 389 surface protein expression level (SPEL) traits (Fig. 2b). Variations in CSF can arise due to altered homeostatic mechanisms that regulate the proportion of a subset within the peripheral blood or mechanisms regulating expression of phenotype-defining markers; variations in the SPEL represent altered mechanisms that regulate expression of a protein at the cell surface, such as signalling, trafficking or promoter/enhancer elements.

Figure 2a illustrates that the level of heritability or environmental influence is different for major leukocyte lineages. Overall, DCs show the highest proportion of traits that are highly

heritable (red); some of this can be ascribed to the strong genetic regulation of the expression of CD32 (ref. 8). In terms of the average heritability of traits, CD4 T cells are next, followed by CD8 cells. B cells, monocytes and the innate-like T cells (γδ and NKT) overall show far less heritability.

Environmental influence can be broken down into unique and shared; the latter identify traits that are correlated among twins due to non-genetic but shared influences. The proportion of variation due to common environmental influences is shown in blue in the circle plots. Inspection of Fig. 2a reveals that certain subsets of CD4 T cells, γδ T cells, NKT and monocytes show the

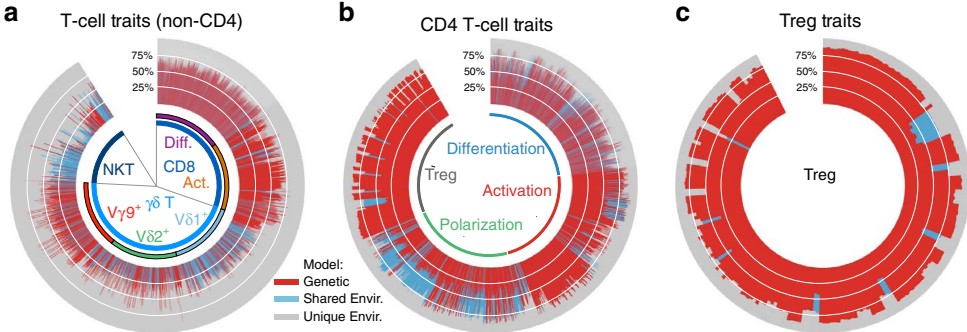

**Figure 3 | Trait variation determination in T cell subsets.** The subset of T cell traits shown in Fig. 2a are expanded. (**a,b**) T-cell traits for non-CD4 T cells (**a**), including CD8 T, γδ T and NKT cells (defined as CD1d-multimer binding), and for CD4 T cells (**b**). Traits were arranged according to the subsets from which they were defined: 'Differentiation'—subsets defined by the markers CD27, CD28, CD31, CD45RA, CD57, CD95, CD127 and CD244 (Staining panel 1; see Supplementary Fig. 2 in Roederer et al.[8]). 'Activation'—subsets defined by the markers CD25, CD38, HLA-DR and CD279/PD-1 (Staining panel 2). 'Polarization'—subsets defined by the markers CCR4, CCR6, CCR10, CXCR3 and CXCR5 (Staining panel 3). 'Treg'—subsets defined by the markers CD25, CD127, CD39, CD45RO and CD73 (Staining panel 2). (**c**) Traits for only Treg cells (expressing CD39 and/or CD73). Of the 20 traits showing shared environmental influence, 18 arise from subsets co-expressing CD25 and CD73 with variable expression of other markers.

highest degree of variation ascribed to shared environmental influences.

**Frequent heritability in adaptive but not innate-like T cells.** The regulation of T-cell immune traits is illustrated in greater detail in Fig. 3. Within the CD4 T-cell population, the highest degree of heritability is within the Treg cells[7,8] (Fig. 3b); this is primarily due to the genetic control of expression (on/off) of the CD39 protein[8]. However, there are also 19 traits in the Treg analysis that show evidence of a significant shared environmental influence (Fig. 3c): these traits include subsets of cells that are CD25⁺CD73⁺ (and variably express other markers, including CD38, CD39, CD45RO, CD127, HLA-DR and PD-1). This suggests that the representation of CD25⁺CD73⁺ Treg, is to a large extent, a consequence of the exposure of individuals to pathogens, diet or microbiome elements, shared in a household during maturation. Given that the twins lived together until the age of 18 years on average and the age of our cohort ranges from 30 to 75 years, it is likely to be that most common environmental influences were imprinted in childhood and are maintained throughout life.

We previously identified a genetic element contributing a small but statistically significant proportion to the representation of CD73⁺ Treg, mapping to an intron within the *CD73* gene itself[8]. This illustrates a relatively common finding: that both genetic and shared environmental influences appear to regulate immune traits.

Other major subsets within CD4 T cells show different degrees of heritability or environmental influence. Among the subsets identified by differentiation stages, there is no clear pattern of association. Subsets identified according to their activation stage (that is, expression of markers such as CD38 and HLA-DR) generally show a higher degree of heritability. This is in direct contrast to subsets identified by the putative degree of functional polarization (based on the expression of chemokine receptors). Within CD4 T cells, these latter subsets uniquely show a high degree of shared environmental influence.

CD8 T cells in general show a similar pattern compared with CD4 T cells (Fig. 3a). Among the innate-like T-cell populations, the Vδ2Vγ9 population (innately programmed to respond to phospholipids) show higher heritability than other γδ subsets. Both the Vδ1 cells, which are among the earliest to differentiate during ontogeny, and NKT cells, which respond to glycolipid

antigens presented in the context of the invariant CD1d, show high degrees of shared environmental influence.

**Environmental influence in B1 cells and cross-presenting APC.** The regulation of non-T-cell traits is illustrated in Fig. 4. Within B cells, there are no traits that are highly genetically regulated; there appears to be more genetic regulation over the naive/immature B cells (IgD) than over the class switched subpopulations. However, there is a striking difference between the B1 and conventional lineages overall: the B1 cell traits show virtually no evidence of genetic influence, but many show significant evidence of shared environmental influence. Thus, the B1 lineage, unlike conventional B cells, is strongly shaped by environmental influences probably during early life.

As noted above, the DC populations generally show a high degree of heritability. The exception to this is the professional antigen presenting cells for CD8 cells (CD11c⁺CD123⁻ CD141⁺), which exhibit shared environmental influence. Finally, monocyte traits overall show a reasonably high degree of shared environmental influence, with relatively lower genetic influence.

**Limited heritability of cell SPELs.** Finally, Fig. 2b illustrates the SEqM results for the SPEL traits. Overall, the pattern is quite different to the CSF traits. In general, for the ~30 proteins that were quantified by our panels, most do not show high degree of heritability. There are some obvious exceptions, such as the expression of CD32 and CD123 on DCs. Interestingly, however, there is a reasonably high degree of common environmental influence on the expression levels of some proteins, such as CD4 (on memory CD4 T subsets) and CD8 (on nearly all CD8 subsets).

**Discussion**

The representation and phenotype of leukocytes in the peripheral blood is generally highly regulated through multiple unknown homeostatic mechanisms. These include both genetically programmed mechanisms to maintain levels of important subsets to achieve a balanced functionality of the immune system, as well as adaptive mechanisms that adjust cell subset levels in response to pathogens, diet, and microbiome. Although the ability of the immune system to generate memory at the antigen specific level (that is, of the T lymphocytes) is well described, it is possible that variation in innate or innate like cells (for example, NKT, MAIT

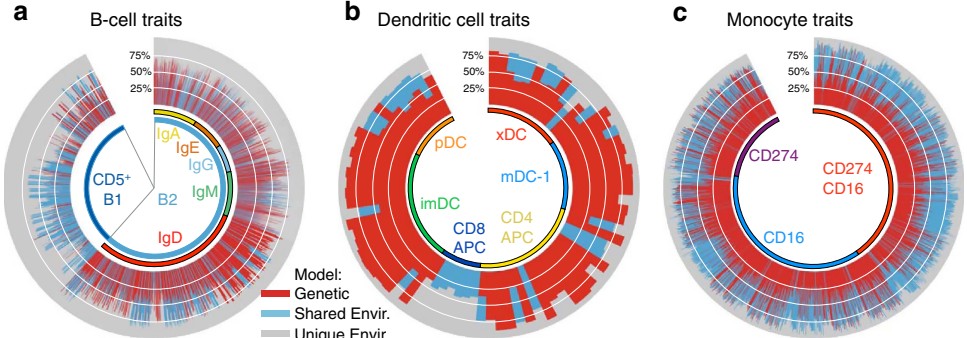

**Figure 4 | Trait variation determination in non-T-cell subsets.** (**a**) B-cell traits were arranged according to lineage (B1 or conventional B2) and among B2, by surface isotype expression (here, cells that co-express IgM and IgD are labelled as IgD; those that express only IgM are labelled as IgM). (**b**) DC traits are arranged by subset phenotype. 'pDC', plasmacytoid DC (CD11c$^-$CD123$^+$). 'xDC', DC co-expressing CD11c and CD123. All others are myeloid DC (mDC, CD11c$^+$CD123$^-$), including mDC-1 (CD1c$^-$CD16$^-$), antigen-presenting cells for CD4 T cells ('CD4 APC', CD1c$^+$), antigen-presenting cells for CD8 T cells ('CD8 APC', CD141$^+$) and inflammatory mDC ('imDC', CD1c$^-$CD16$^+$). (**c**) Monocyte traits are divided by the coordinate expression of CD16 and CD274.

and γδ T cells) is more influenced by exposure to infections, microbes and other immunological insults.

We chose to use cryopreserved peripheral blood mononuclear cell (PBMC) for our study. This affords the many advantages of batch analysis (for example, all co-twins were analysed in the same run; only 30 runs were required for the nearly 800 samples). Strict quality control (QC) during preservation and thawing ensured that all samples passed viability thresholds. Nonetheless, there are disadvantages of cryopreserved specimens—some cell markers are labile to the processing and could not be assessed. Experimental error arising from methodological aspects can contribute to unique environmental influences in genetic studies and so lower heritability estimates, but specifically do not contribute to shared environmental influences. Notably, Orru et al.[7], who used fresh PBMC, did not observe higher heritabilities than we did, indicating no substantial benefit of fresh analysis.

One of the goals of our study was to define potential influences on homeostasis—that is, how the body regulates numbers of cells in the peripheral blood. It is reasonable to postulate that such mechanisms operate through recognition of cell types based on cell surface marker expression. Although the correlation between surface phenotype and functions are manifold, is quite possible that there will be significant associations between genetics or environment and functionally defined subsets of leukocytes. Certainly, future studies can include functional studies to elucidate such relationships. In addition, our analysis has been limited to the cells in the peripheral blood for pragmatic reasons. Homeostatic mechanisms controlling tissue levels of leukocytes as well as homing mechanisms remain to be addressed and are unlikely to be elucidated from our data.

The impact of heritability on the human immune cell homeostasis is just beginning to be understood. Twin studies have been particularly powerful at revealing the scale at which the immune system is genetically controlled. Two major studies[6,7] recently focussed on seemingly conflicting messages relating to the heritability of immune traits. Orru et al.[7], using a large number of Sardinian families, reported that the majority of the analysed cell types are genetically driven. In contrast, Brodin et al.[6], using 210 (mostly MZ) twin pairs and combining immunophenotyping data from distinct platforms, concluded that variation in the immune system is mostly driven by environmental or experimental factors. In our previous report[8], using 245 twin pairs, a single immune phenotyping platform and independent validation using another 164 twin pairs, we focused on identifying a number of genetically associated traits by evaluating the top 151 heritable traits based on our twins analysis.

Here we focus on a global and comprehensive analysis of 23,394 immune phenotypes in 497 twins, to determine the overall contribution of genetic versus heritable influences. Many traits show evidence of both genetic and environmental factor contributions. Nonetheless, the majority (76%) of the analysed immunological traits are predominantly genetically controlled and a considerable number of immune traits, which are under shared environmental but not genetic control.

In addition, we took advantage of the twin structure of our cohort to model the common environmental influence on the same traits. This analysis allows us to ascribe the proportion of the variation of any given trait that is explained by heritable or genetic influences, shared common environmental influences, or by unshared environmental influences or simply stochastic in nature[9]. Our results showed that the best fitting model for the majority of the analysed immune cells includes genetic factors, but that shared environmental influences are often apparent.

There are some limitations of our data. For historical reasons, the majority (85%) of the TwinsUK cohort included females and in this study we have analysed females only. It is possible (although in our view unlikely) that some of the results may differ in males. Although we had a reasonable sample size, very large numbers are needed to exclude small effect of shared environment[10]. We had, however, good power to show heritabilities above 25% (ref. 11).

The relatively homogeneous composition of our cohort afforded us power to detect associations that may have been lost in a much more heterogeneous cohort and serves as a proof-of-principle that such studies can elucidate important gene and environment influences on the peripheral immune system. Extending this study by including greater numbers will help identify more gene loci and adding diverse populations will undoubtedly reveal more interactions with the environment, as these analyses are population specific. Having more longitudinal data would have improved our estimates of error and random variation, and although our samples were cryopreserved we believe the results were robust based on our replication samples and any error due to differential changes would have reduced heritability estimates not falsely inflated them.

Our data show that certain subsets within leukocytes are more likely to show shared environmental influences (Fig. 5). Examination of these trends reveals common threads. For example, the predilection towards a shared environmental influence of CD4 polarization markers (for example, Th1 versus Th2 polarization), NKT cells, Vδ1T cells and B1 cells is likely to be due to a shared exposure to microbial products during childhood/adolescence:

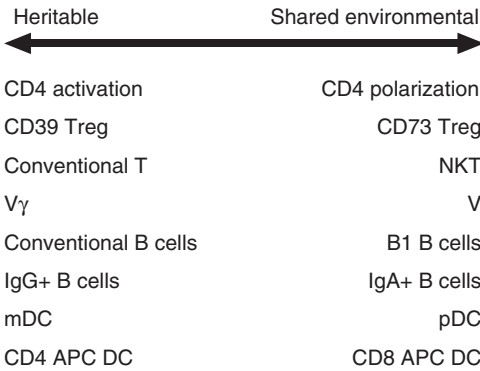

| Heritable | Shared environmental |
|---|---|
| CD4 activation | CD4 polarization |
| CD39 Treg | CD73 Treg |
| Conventional T | NKT |
| Vγ | V |
| Conventional B cells | B1 B cells |
| IgG+ B cells | IgA+ B cells |
| mDC | pDC |
| CD4 APC DC | CD8 APC DC |

**Figure 5 | Influences on leukocyte subset representation.** SEqM quantifies the degree of variation in traits that can be ascribed to heritable versus shared environmental influences. Here are illustrated divergent trends in major subsets. For each comparison, the illustration indicates a relative increase in either heritable or shared environmental influences. For example, subsets of CD4 T cells defined by polarization markers (chemokine receptor expression) show increased shared environmental influences compared to C4 subsets defined by activation markers (CD38, HLA-DR).

either through microbiome or common infections. Many of these cell types bridge the span between the innate and the adaptive immune system[12] and have been postulated to have functional roles in the immunity to common bacterial or parasitic infections. However, although their representation is partially genetically programmed through innate mechanisms, the actual exposure to particular pathogens during adolescence may imprint a lifelong change in the representation of those cells to provide a long-term memory of a different type than the adaptive B- and T-cell antigen-specific cells.

Those subsets with unusually high heritable or shared environmental influences may represent rich targets for further research. Understanding the mechanisms by which pathogens, diet or microbiota imprint long-lasting changes in the immune system will be important for the design and implementation of precision medicine interventions that may be influenced by these subsets. A proof of principle of this concept was the demonstration that genetic elements previously shown to predispose towards autoimmunity have profound impact on the phenotype of DCs[8].

In summary, we analysed the genetic and environmental influences on over 23,000 immune traits, covering the representation and phenotype of all major lymphocyte and myeloid subsets in the peripheral blood. We find genetic influences on a large fraction of these traits. In addition, we identify classes of leukocytes that exhibit significant shared environmental influences, indicating an imprinting from dietary, healthy microbial or pathogen exposures that occurred between birth and adolescence and is maintained through adulthood. Our data provide insight into the homeostatic mechanisms governing the balance of immune cells in the peripheral blood, which ultimately impact immunological function, both protective and autoimmune.

## Methods

**Human data.** This study was approved by the NIAID (NIH) IRB and London-Westminster NHS Research Ethics Committee; all participants provided informed consent. The UK Adult Twin Register, TwinsUK and the immunophenotyping methods have been previously described in detail[8,13]. Briefly, TwinsUK is a large cohort of twins historically developed to study the heritability and genetics of diseases with a higher prevalence among women. The population is not enriched for any particular disease or trait and is representative of the British general population[14]. Zygosity of twin pairs was determined by genome-wide genotyping[8] or using 16 short tandem repeat DNA markers. For the analysis here, we used data

from 75 pairs of MZ ($n = 150$ individuals) and 170 pairs of DZ ($n = 340$ individuals) twins and 7 singletons (arising from QC failures in one co-twin) who were aged from 41 to 77 years old (the 'discovery' cohort fully detailed in Roederer et al.[8]).

For reproducibility analysis, 29 healthy controls from the United States were included in the study, from whom blood was drawn and cryopreserved at two time points at least 6 months apart. An additional (replicate) vial from one of the two blood draws was analysed for 14 of 29 of these controls. The samples in each experimental staining day were ordered such that twin or longitudinal control samples were analysed in the same experimental run (each comprising 15–30 vials), whereas replicate control samples were analysed in different experimental runs. Staining and data analyses were otherwise performed blinded to identity.

**Immune trait quantification.** A full description of the quantification of immune cells and their phenotype is described elsewhere[8]. All samples were cryopreserved within 6 h of collection and maintained on liquid nitrogen until processing. All samples passed QC criteria for cryopreservation: >70% viability on thaw and only live cells included in the analysis by virtue of inclusion of a live/dead liability marker. Although the use of cryopreserved PBMC introduces some limitations to the phenotyping, it has the distinct advantage of eliminating run-to-run variability: all co-twin samples were analysed in the same run and batch variation is therefore significantly reduced. Immunophenotyping panels are summarized in Supplementary Fig. 1.

**Selection of traits and modelling influences.** In our previous report[8], only the most heritable traits (as identified by the Falconer's approximation) were included for further analysis by GWAS; we analysed 151 traits selected from a total of 78,683 traits. In the interim, a different gating strategy was applied to the B-cell panel (Supplementary Fig. 1), after which we replaced 567 original B-cell traits with a more comprehensive set of 10,935 B-cell traits (Supplementary Fig. 2), for a grand total of 89,051 traits.

These 89,051 measurements comprise roughly 1,500 fully independent traits (and many more partially independent)[8]. However, it is impossible to define which of them is the predicate trait for any given family of highly correlated values. Thus, we set out to analyse all as potential traits. Any given phenotype of cell population of interest is thus represented within the totality of the data and can be interrogated for its dependence on heritability or shared environmental influences.

Here we performed SEqM on all robust traits. Several criteria were applied to determine whether a trait was robust (Supplementary Fig. 3): for frequency traits (that is, CSF), the trait value (proportion of a given cell subset within its parent lineage) had to be >0.1 and <99%. For phenotype traits (that is, the SPEL of a marker on cell subsets), the median fluorescence intensity had to be >100. This criterion eliminated 32,124 traits. For representation and phenotype traits, the reproducibility among the longitudinal specimen values had to be >0.7 to be considered robust. This criterion, which filters out traits that are highly dynamic over time and/or experimentally hypervariable, eliminated 18,930 traits. Finally, for simplification, we eliminated traits derived from cell lineages that are poorly described in the literature, such as CD4$^+$CD8$^+$ (double positive) or CD4$^-$CD8$^-$ (double negative) T cells. This eliminated 14,603 traits.

**Statistics.** To estimate the heritability of the different immune cell types, we used SEqM. This standard method for twin studies quantifies sources of individual variation by decomposing the total phenotypic variance into genetic and environmental variances[15]. The environmental variance component can be further subdivided into a common/shared (C) environmental component (representing environmental factors affecting both twins in a pair, and a source of similarity) and an individual/unique environmental component (E) (environmental factors acting differently in the twins of a pair and making them dissimilar). E also contains measurement error. In particular, we used the ACE model, with A representing the additive genetic effects, C the shared and E the non-shared environmental effects. Comparisons on performances between the full ACE model and its nested models (AE, CE and E) were performed using the likelihood ratio test. The likelihood ratio test calculates twice the difference in the log likelihoods for the full and the nested models and can be approximated by a $\chi^2$-distribution with degree of freedom equalling the difference in the number of parameters in the two models. Selection of the best-fitting submodel was based on a balance between goodness of fit and parsimony[11] The Akaike information criterion (AIC) is a measure of the goodness of fit of a statistical model[15]. AIC describes the balance between accuracy and complexity of the models, and therefore provides a means for model selection based on the lowest AIC value. Parsimony is a non-parametric statistical method where the evolutionary model that has the highest probability of producing the observed data is the most likely model. Estimation of heritability, and the best-fitting model were performed using the openMX software[16].

It is noteworthy that the selection of the best-fitting model using likelihood ratio test (LRT) and AIC (and the degree of freedom for each trait) is performed independently for each trait and, therefore, the final interpretation of these results is not affected by the number of test conducted.

Estimates of heritability can be biased if greater sharing of environmental confounders strongly related to the trait occurs more commonly in the MZ pairs

compared with DZ pairs. All twins included in this study were raised together and shared similar degree of socio economic status (86% of the twin couples had similar education level and 89% of them live in a similar urban area indicating a comparable welfare). Moreover, there was no statistical difference in both education level and welfare between MZ and DZ. As previous studies reported that immune traits are associated with age[4,7] and, based on the fact that in our cohort MZ pairs were on average 2 years older than DZ pairs, the heritability estimates of each trait were adjusted for age by linear regression after removing the outliers ($>4$ s.d.). The residuals were then normalized using inverse normal transformation and used for the heritability analysis. We then selected 100 random traits to check that the distribution of the traits was normal in both the whole data set and the twin subgroups (MZ and DZ).

**Data availability.** Genotype data are available upon request to the authors. Data for flow cytometry, twin demographics and trait values were previously described in ref. 8 and are accessible using the following depositories. Anonymized demographic information is provided in a single file ftp://twinr-ftp.kcl.ac.uk/ImmuneCellScience and measured trait values can be downloaded from ftp://twinr-ftp.kcl.ac.uk/ImmuneCellScience. Flow cytometry data and FlowJo workspaces are deposited in the International Society for the Advancement of Cytometry public data repository.

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

## Acknowledgements

We thank Isabella Tosi, Zeynep Catak, Leena Khatri, Luca Napolitano, Manuela Terranova Barberio, Federica Villanova, Paola Di Meglio, Gabriela Surdulescu, Dylan Hodgkiss and Ayrun Nessa for technical help with the samples; Pratip Chattopadhyay, Yolanda Mahnke, Kairmei Song and other members of the ImmunoTechnology section for technical assistance and advice; and the TwinsUK volunteers for participation. This work was supported by the Vaccine Research Center (NIAID, NIH) Intramural Research Program and by the Department of Health via the National Institute for Health Research (NIHR) comprehensive Biomedical Research Centre award to Guy's & St Thomas' NHS Foundation Trust in partnership with King's College London and King's College Hospital NHS Foundation (guysbrc-2012-1) Trust, and Dunhill Medical Trust. TwinsUK is also supported by the Wellcome Trust and T.D.S. is an NIHR Senior Investigator.

## Author contributions

M.R., T.D.S. and F.O.N. designed and supervised the study. M.H.B. collected and analysed data. M.M. and M.R. performed data analysis. M.M. performed statistical analyses. M.M., M.R., T.D.S. and F.O.N. wrote the manuscript.

## Additional information

**Competing financial interests:** The authors declare no competing financial interests.

