## [Peer Review File · Nature Communications]

Reviewers' comments:

Reviewer #1 (Remarks to the Author):

The manuscript by Mangino and colleagues investigates the differential impact of genes and environment on the numbers and cell markers expression of various immune cells in a large cohort of female twins. The authors should be commended for the excellent effort and for the important information provided by their study. Although the efforts and importance of the study are obvious, I think that the interpretation of the findings should be toned down, especially at the level of terminology, as explained below.

Comments:

1. To this reviewer it is unclear what do the authors mean by 'immune traits'? 7 panels of 13-14 molecules have been used, but how does that translate in a 'grand total' of some 89.000 grant traits (or 23.000 in another count)? Are these combinations of expression of a couple of different molecules? If so, what does that mean, do all these "traits" have any physiological relevance? This is doubtful, and also misleading to call them 'immune traits'. From a functional point of view, an immune trait would be the capacity to present an antigen, or to phagocytose a pathogen, or indeed the number of B-cells, but not each spurious combination of molecules expressed on cell surface. From my perspective this terminology should be changed.
2. While the authors should be commended for the breadth of the study, they should also acknowledge that this study assesses cell numbers and marker expression and no more than that. If we would want to investigate for example the broad impact of genes and environment on DCs, both numbers and function should be considered. The numbers may be identical in two individuals, but the capacity to phagocytose or present antigens is not only dependent on the expression of cell membrane molecules involved in these processes.
3. Therefore, the conclusions should be more modest and focused on cell numbers, and no broad assumption regarding the function of the immune system or of the individual cell populations can be reliably made solely based on expression patterns. Physiological assays would be needed for that: for example monocyte function should be assessed by cytokine stimulation assays and cannot be given by expression of CD14 or CD16 (indeed that is not reliable at all), etc. This limitation should be acknowledged in the title, abstract and discussion.
4. Discussion is short and lacks some important points. The Discussion should also point out that the patterns observed here are for blood cell populations, and different conclusions may be true for cell populations in the various tissues.
5. Can the authors discuss whether they would expect a similar pattern of gene-environment interaction in other population of different genetic background and/or different infectious pressure?

Reviewer #2 (Remarks to the Author):

This study is an extension of Roederer et al Cell 2015, as a "Resource" paper (ref 8): may more flow cytometric phenotypes and estimating genetic versus environmental influences. This is a better and more up to date resource. In the 2015 they also analysed the genetic associations with 151 of these PBMC traits, but for some reason, here, they do not expand this number...it still looks as if it's 151 from the Methods. Can they please analyse more than 151 with the expanded dataset? It might be of value to connect more immune trait SNPs with GWAS disease SNPs, as they published previously, and before them, Orru V et al 2013 (ref 7). In fact, in Roederer et al the main novel genetic connections were not that new over and above Orru et 2013, and hence why the paper was published as a "Resource". This was despite the deeper FACS phenotyping by Roederer.

So, one major justification for the study is that there is uncertainty as to the relative contribution of genes versus environment for these blood traits and Mangino et al set to resolve this "controversy". I don't think it's that controversial. Roederer et al and Orru et al found that genetics was more contributory than environment whereas Brodin et al Cell 2015, claimed environment predominated. When Brodin et al published I did wonder if they chose the title to help justify publication. Inspection of the results across studies suggests that the actual differences aren't that different, it's how to want to place the emphasis. The most telling result in Brodin et al is when they admit that in younger twins many more traits show heritability. So, if twins, under age 20 yr, were analysed in sufficient numbers and in phenotypic depth, everyone would be reaching the same conclusions. Its ageing and long-term environmental exposures that reduce the detectable heritability. In the first years of life there is considerable genetic influence...but, so what? Do these genetic associations lead to better understanding of GWAS associations.....CD39 didn't, and as above, what's the new information for traits over the original 151? Further, having phenotype association data and disease association data (GWAS) can help in fine mapping a region of association - has this been attempted? They seem to only remind us of the well-studied CD32/FCGR2A and CD39 results, but what are the main gene specific "novel findings"? They cite Carr et al (ref 4) as a Resource paper in Nat Immunol in 2016 as age and cohabitation as being major factors in determining blood phenotypes - they should discuss these results more. They should mention the limitations of frozen PBMCs.....many surface receptors are lost or greatly reduced on freezing e.g. the CCRs, and the degree to which this happens, length of storage time, operator variation, reagent and equipment variation. This could be a considerable environmental influence, particularly on highly reactive cells such as monocytes, or cell that are very sensitive to freeze-thaw (i.e. variable numbers die). If the PBMCs were stored for a long time, if the shipping was not on liquid nitrogen, if the freezing and thawing protocols were not optimised, then large of portions of heritability could be lost. Note that Orru et al used fresh blood, not PBMCs, and hence avoided the loss of cells and activation and temperature shock caused by the PBMC process.

Reviewer #3 (Remarks to the Author):

The main method of SEM to analyze twin data, LRT and AIC as the model selection criterion, are established and valid methods. This is a very interesting study attempting to determine heritability of tens of K phenotypes using the dataset with 100s degrees of freedom (twins). While the findings are interesting and worthwhile, just how many independent inferences these authors can make on their materials remain unclear.

This study is original, but also it is an extension of the previous effort; just for large number of phenotypes, thus innovation is on lower end.

Additional smaller comments are below:

1. Were all the twin pairs raised together? If not, the shared environment component will not apply to all pairs.
2. Material and Methods: "Estimates of heritability can be biased if greater sharing of ...": If information is available on SES (socio-economic status) or occupation of subjects, it will provide an at least qualitative answer to the question of similarity or dissimilarity of adult life environment between MZ and DZ twins. Is such info available?
3. In this SEM, error cannot be separated from unique environmental effect directly. Is it possible to use the variance from longitudinal repeats to estimate error and take it out of environmental variance completely?
4. Discussion: "Two major studies recently focused on conflicting messages relating to the heritability of immune traits": Heritability being a population-level measure and not merely a property of the studied trait, discuss how the baseline genetic diversity or homogeneity and environmental variability or similarity (both natural geography and health-care history e.g. immunization plans) across the subject panel of each study might affect estimate of heritability.

AU: Reviewer comments are included in their entirety and prefaced with “REVn”; our responses follow and are prefaced with “AU”.

Reviewer #1 (Remarks to the Author):

REV1: The manuscript by Mangino and colleagues investigates the differential impact of genes and environment on the numbers and cell markers expression of various immune cells in a large cohort of female twins. The authors should be commended for the excellent effort and for the important information provided by their study. Although the efforts and importance of the study are obvious, I think that the interpretation of the findings should be toned down, especially at the level of terminology, as explained below.

AU: We appreciate the comments – see below regarding terminology. We have modified our discussion to temper the interpretations, and as requested by other reviewers, added some clarifications as well.

REV1: 1. To this reviewer it is unclear what do the authors mean by 'immune traits'? 7 panels of 13-14 molecules have been used, but how does that translate in a 'grand total' of some 89,000 grant traits (or 23,000 in another count)? Are these combinations of expression of a couple of different molecules? If so, what does that mean, do all these "traits" have any physiological relevance? This is doubtful, and also misleading to call them 'immune traits'. From a functional point of view, an immune trait would be the capacity to present an antigen, or to phagocytose a pathogen, or indeed the number of B-cells, but not each spurious combination of molecules expressed on cell surface. From my perspective this terminology should be changed.

AU: The traits were defined in our original Cell paper. Traits are indeed defined based on the combination of expression of between 4 and 14 markers. And, we fully agree that many of these traits may have no physiological relevance. However, we do not know *which* do, and *which do not*! This is a key part of our in-depth analysis – rather than taking a biased and likely incomplete definition of a given subset of cells based on literature or limited functional analysis, we looked at all possible combinations. As assessed in the Cell paper, our 89,000 traits likely comprise only about 1,500 completely independent measurements, with many more partially independent. There is no way to currently define which traits are the “key” functional ones. Our hypothesis was that those traits which are genetically or environmentally influenced would include those fundamental subsets (as well as correlated “spurious” subsets). Our statistical analysis corrected for multiple comparisons using the far more strict 89,000 measurements.

Thus, while we agree that there are not 89,000 traits in our dataspace, we assert

that there are conservatively at least 1,500. We could refer to the larger set as “potential immune traits” but think this might be confusing. We added discussion to the manuscript to introduce and clarify these points. Certainly, conclusions drawn on the full set of measurements will be valid for the predicate independent subset of traits.

REV1: 2. While the authors should be commended for the breadth of the study, they should also acknowledge that this study assesses cell numbers and marker expression and no more than that. If we would want to investigate for example the broad impact of genes and environment on DCs, both numbers and function should be considered. The numbers may be identical in two individuals, but the capacity to phagocytose or present antigens is not only dependent on the expression of cell membrane molecules involved in these processes.

REV1: 3. Therefore, the conclusions should be more modest and focused on cell numbers, and no broad assumption regarding the function of the immune system or of the individual cell populations can be reliably made solely based on expression patterns. Physiological assays would be needed for that: for example monocyte function should be assessed by cytokine stimulation assays and cannot be given by expression of CD14 or CD16 (indeed that is not reliable at all), etc. This limitation should be acknowledged in the title, abstract and discussion.

AU: The reviewer is correct that no functional measures were performed; however, there is ample precedent for associating phenotype with certain functions. (E.g., CD25+CD45RO+CD127- CD4 T cells having regulatory function; CD45RO+CD28+CCR7+ T cells providing long-term recirculating memory; etc.). Future studies such as this may well focus on in vitro functional correlates.

One of our original goals in the study was to elucidate mechanisms of homeostasis (i.e., whether they were genetically or environmentally signaled; and if the former, which gene loci contributed). It is reasonable to postulate that mechanisms regulating homeostatic levels of cell populations do not depend on the functionality of the cells themselves; but rather, are mediated by and through recognition of cell surface markers (i.e., counting the cells in the body).

We note that there are multiple clinical associations with phenotypes or with alleles however; part of the goal of our database is to provide a bridge between these disparate findings, and the goal of the present paper to put the former in some context. One of the powers of our study is to provide researchers with the ability to estimate heritability and/or shared environmental influence on any given leukocyte subset that they might be researching. The breadth of our analysis provides the greatest chance that a subset of cells of interest are included.

REV1: 4. Discussion is short and lacks some important points. The Discussion should also point out that the patterns observed here are for blood cell populations, and different conclusions may be true for cell populations in the various tissues.

AU: These important caveats have now been added.

REV1: 5. Can the authors discuss whether they would expect a similar pattern of gene-environment interaction in other population of different genetic background and/or different infectious pressure?

AU: We certainly expect so, and are gearing up for such a study. The limitations (and, incidentally, advantages) of using our homogeneous cohort were added to the discussion.

Reviewer #2 (Remarks to the Author):

REV2: This study is an extension of Roederer et al Cell 2015, as a "Resource" paper (ref 8): may more flow cytometric phenotypes and estimating genetic versus environmental influences. This is a better and more up to date resource. In the 2015 they also analysed the genetic associations with 151 of these PBMC traits, but for some reason, here, they do not expand this number...it still looks as if it's 151 from the Methods. Can they please analyse more than 151 with the expanded dataset? It might be of value to connect more immune trait SNPs with GWAS disease SNPs, as they published previously, and before them, Orru V et al 2013 (ref 7). In fact, in Roederer et al the main novel genetic connections were not that new over and above Orru et 2013, and hence why the paper was published as a "Resource". This was despite the deeper FACS phenotyping by Roederer.

AU: The present study indeed includes all (robust) traits in our data base; Figure S3 illustrates the trait selection process. Our Cell paper was limited to 151 per a rigorous pre-specified statistical analysis plan to account for multiple testing. The limit was lifted for this analysis. As for the GWAS correlations with all traits: that process is ongoing and we hope to begin analyzing the results soon. However, it is both computationally and temporally expensive to perform and interpret ~23,000 separate GWAS analyses.

REV2: So, one major justification for the study is that there is uncertainty as to the relative contribution of genes versus environment for these blood traits and Mangino et al set to resolve this "controversy". I don't think it's that controversial. Roederer et al and Orru et al found that genetics was more contributory than

environment whereas Brodin et al Cell 2015, claimed environment predominated. When Brodin et al published I did wonder if they chose the title to help justify publication. Inspection of the results across studies suggests that the actual differences aren't that different, it's how to want to place the emphasis. The most telling result in Brodin et al is when they admit that in younger twins many more traits show heritability. So, if twins, under age 20 yr, were analysed in sufficient numbers and in phenotypic depth, everyone would be reaching the same conclusions. Its ageing and long-term environmental exposures that reduce the detectable heritability. In the first years of life there is considerable genetic influence...but, so what? Do these genetic associations lead to better understanding of GWAS associations.....CD39 didn't, and as above, what's the new information for traits over the original 151? Further, having phenotype association data and disease association data (GWAS) can help in fine mapping a region of association - has this been attempted? They seem to only remind us of the well-studied CD32/FCGR2A and CD39 results, but what are the main gene specific "novel findings"?

AU: We fully agree with the Reviewer's analysis – unfortunately, based on others researchers' public presentations at meetings (and on citations), there still appears to be some confusion (controversy) arising from interpretations presented in Brodin. Note that Orru were unable to analyze shared environmental impact because of the study design.

Note that in this paper we focus more on that aspect: the impact of shared environment on traits – than on specific gene contributions. As such, this is the most detailed analysis of environmental impact on the immune system to date.

REV2: They cite Carr et al (ref 4) as a Resource paper in Nat Immunol in 2016 as age and cohabitation as being major factors in determining blood phenotypes - they should discuss these results more. They should mention the limitations of frozen PBMCs.....many surface receptors are lost or greatly reduced on freezing e.g. the CCRs, and the degree to which this happens, length of storage time, operator variation, reagent and equipment variation. This could be a considerable environmental influence, particularly on highly reactive cells such as monocytes, or cell that are very sensitive to freeze-thaw (i.e. variable numbers die). If the PBMCs were stored for a long time, if the shipping was not on liquid nitrogen, if the freezing and thawing protocols were not optimised, then large of portions of heritability could be lost. Note that Orru et al used fresh blood, not PBMCs, and hence avoided the loss of cells and activation and temperature shock caused by the PBMC process.

AU: Our panels were specifically optimized to avoid freeze/thaw "artefacts" such as receptor expression problems (e.g., some of our panels used 37C staining; we did not evaluate CD62L on T cells; we avoided CCRs that are labile).

As noted in our original paper, all samples were carefully cryopreserved and kept on LN2 for the duration (less than 2 years). Cryopreservation was performed by following an SOP optimized in our preclinical laboratories. In addition, all samples were quality controlled for viability upon thaw, and all passed our criterion for analysis (>60% viable on thaw). In addition, all panels included a live/dead viability marker, and all phenotyping was performed on live cells only.

We also analyzed replicate samples and longitudinal samples to assess experimental variability. Notably, the entire study was performed by one operator on one machine. Traits which showed poor replicability (<70% covariance across three samples taken over a six month timeframe from the same 30 control individuals) were not considered in our analysis (see figure S3).

There are clearly disadvantages to using cryopreserved PBMC; there are also advantages. A major advantage is the ability to analyze in batch to eliminate day-to-day experimental variability (i.e., all co-twin samples were analyzed in the same run). In any case, the limitations (and, indeed, advantages) of cryopreserved vs fresh PBMC was added as a discussion point.

Finally, we note that experimental error arising from methodological aspects like these can contribute to unique environmental influences in genetic studies and so lower heritability estimates, but specifically do not contribute to shared environmental influences. Notably, if the Orru et al method was more robust than ours, they would have seen higher heritabilities than we did – which was not the case.

Finally, a significant power of our twin study is the ability to specifically isolate shared environmental influences on immunological traits.

We have now added discussion of these points to the manuscript.

Reviewer #3 (Remarks to the Author):

REV3: The main method of SEM to analyze twin data, LRT and AIC as the model selection criterion, are established and valid methods. This is a very interesting study attempting to determine heritability of tens of K phenotypes using the dataset with 100s degrees of freedom (twins). While the findings are interesting and worthwhile, just how many independent inferences these authors can make on their materials remain unclear.

This study is original, but also it is an extension of the previous effort; just for large number of phenotypes, thus innovation is on lower end.

AU: we respectfully disagree; the novel contribution here is the identification of shared environmental influences on a wide range of immunological traits. This is something that is uniquely afforded by this balanced twin cohort study.

We estimate that there are about 1,500 clearly independent traits (please see reviewer 1 Q1). However, we would like to clarify that the selection of the best fitting model using LRT and AIC (and the degree of freedom for each trait) is performed independently for each trait and, therefore, the final interpretation of the results is not affected by the number of test conducted.

REV3: 1. Were all the twin pairs raised together? If not, the shared environment component will not apply to all pairs.

AU: All twins included in this study were raised together. A sentence to clarify this point has been added in the material and method and results sections.

REV3: 2. Material and Methods: "Estimates of heritability can be biased if greater sharing of ...": If information is available on SES (socio-economic status) or occupation of subjects, it will provide an at least qualitative answer to the question of similarity or dissimilarity of adult life environment between MZ and DZ twins. Is such info available?

AU: As suggested by the reviewer, to further examine the twins' adult life environmental similarity/dissimilarity, we analyzed two main socio-economic factors: twins' education (considering the different level of education achieved by co-twins) and their wealth (based on the UK regional index of welfare, we analyzed co twin differences using their area post codes). The SES analyses revealed that: a) 86% of the twin couples have a similar level of education (both co-twins finished high school and went to university); b) in 89% of the cases both co-twins live in an area with similar index of deprivations indicating that they share the same or similar welfare. Moreover, when comparing MZ versus DZ pairs we did not observed any statistical significant difference in either education level ($p=0.09$) or personal welfare (0.2).

Overall both analyses showed that, in their adult life, twins included in this study share high a level of environmental similarities. This is consistent with the equal environmental assumption (EEA) which underlies any classical twin study. One sentence to clarify this point has been added to the manuscript in both the material and method and in the result sections.

REV3: 3. In this SEM, error cannot be separated from unique environmental effect directly. Is it possible to use the variance from longitudinal repeats to estimate error and take it out of environmental variance completely?

AU: We agree that would be ideal. Unfortunately we do not have longitudinal measurements on the twins themselves. However, we did make longitudinal and replicate measurements on a set of 30 healthy controls, and have a limited estimate of the experimental/longitudinal variability of the immunological traits. Indeed, we used this analysis to prune those traits with limited replicability from our analysis. Early on in our analysis, we considered using this variance to estimate unique environmental variance, but feel that the precision is too poor (given that replicability is estimated on only 30 individuals) and would make this analysis too speculative.

REV3: 4. Discussion: "Two major studies recently focused on conflicting messages relating to the heritability of immune traits": Heritability being a population-level measure and not merely a property of the studied trait, discuss how the baseline genetic diversity or homogeneity and environmental variability or similarity (both natural geography and health-care history e.g. immunization plans) across the subject panel of each study might affect estimate of heritability.

AU: Please see answer to reviewer 1, Q5.

REVIEWERS' COMMENTS:

Reviewer #2 (Remarks to the Author):

Immune characteristics are a combination of genetics and environment....as I said, I don't see the controversy. The data would be value deposited and accessible.